# Health providers readiness in managing intimate partner violence in public health institutions, Ethiopia

Lidiya Teshome[1], Haweni Adugna[2], Leul Deribe[2]*

1 Midwifery Department, Hawassa Health Science College, Hawassa, Ethiopia, 2 School of Nursing and Midwifery, College of Health Sciences, Addis Ababa University, Addis Ababa, Ethiopia

* leul.deribe@gmail.com

## Abstract

### Introduction

Intimate Partner Violence (IPV) is a worldwide public health problem and major human and legal rights abuses of women. It affects the physical, sexual, and psychological aspects of the victims therefore, it requires complex and multifaceted interventions. Health providers are responsible for providing essential healthcare services for IPV victims. However, there is a lack of detailed information on whether or not health providers are ready to identify and manage IPV. Therefore, this study aimed to assess health providers' readiness and associated factors in managing IPV in public health institutions at Hawassa, Ethiopia.

### Method

Institutional based cross-sectional study was conducted through a simple random sample of 424 health providers. Data was collected with an anonymous questioners using physician Readiness to Manage Intimate Partner Violence Survey (PREMIS) tool. Linear regression analysis was used to examine relationships among variables. The strength of association was assessed by using unstandardized β with 95% CI.

### Results

The mean score of perceived provider's readiness in managing IPV was 26.18± 6.69. Higher providers age and providers perceived knowledge had positive association with provider perceived readiness in managing IPV. Whereas not had IPV training, absence of a protocol for dealing with IPV management, and provider attitude had a negative association with provider perceived readiness in managing IPV.

### Conclusion and recommendation

This study reviled that health providers had limited perceived readiness to manage IPV. Provision of training for providers and develop protocol for IPV managements have an important role to improve providers readiness in the managements of IPV.

**Data Availability Statement:** The data set analyzed during the current study was attached as Supporting information files.

**Funding:** Addis Ababa University will provide fixed amount of financial support for all post graduate

students. Accordingly as this research work was part of post graduate theses for primary author, Lidiya Teshome, we received 25,000 ETB from the university. The support has no any Grant numbers. The funders had no role in study design, data collection and analysis, decision to publish, or preparation of the manuscript.

**Competing interests:** The authors have declared that no competing interests exist.

**Abbreviations:** IPV, Intimate partner violence; PREMIS, Physician readiness to manage intimate partner violence survey tool; VIF, Variance inflation factor.

## Introduction

Intimate partner violence (IPV) is a behavior in an intimate relationship that causes physical, sexual, or psychological harm [1, 2]. It occurs regardless of cultural, economic, and religious background, also, women and children are extremely affected by it [3, 4]. Globally one in every three women have experienced physical and/or sexual or both IPV types in their lifetime and the prevalence is high in developing countries [5]. Sub-Saharan Africans are severely affected region in the world with lifetime IPV of 37% [6, 7]. In Ethiopia, IPV is common in both urban and rural areas and it is often not disclosed due to family silence, cultural norms, fear, and shame [8]. About 30.2% of ever-married women in Ethiopia experience one type of IPV in their lifetime and 33.5% to 41.6% women reported IPV during pregnancy [9, 10].

About 15% to 71% of victims develop gynecological, central nervous system, and stress-related problems [9]. In addition, it is a reason for 40–70% of global female homicides [3, 4]. Medical treatments and supports are essential to reduce these adverse medical and social consequences of IPV [11]. Moreover, providing effective care and maintaining appropriate health providers' redness are vital to treat victims and prevent further consequences [12]. Early identification and treatment of physical, sexual, and mental impact of IPV were not sufficiently addressed by the health systems and the health providers were reluctant to provide comprehensive care [13]. Therefore, the victims often prefer to visit religious leaders, relatives, and friends for seeking care and support due to inadequacy of the care they obtain from health facilities [14].

However, Health providers are responsible for providing essential health care services to IPV victims [15]. Therefore, they should have adequate readiness in terms of knowledge, attitudes, and skills [14]. Indeed, the quality of health care for women who experienced violence were poor and majority of health care professionals are unclear about their role in the responses of IPV and they lack basic knowledge and skill in IPV management. Furthermore, they perceive as it is difficult to recognize victims and help women to disclose their experiences [16, 17].

Study conducted in Italy showed limited knowledge of the most common signs and symptoms of violence, reserved the provider from the identification of violence [18]. On the other hand, a study conducted in Egypt and Kenya confirmed lack of skills to communicate prohibits care delivery for IPV victims [19, 20]. A study done in Nigeria also revealed the level of perceived provider knowledge about IPV significantly affects perceived preparedness regarding IPV management [17].

Indeed, previous studies have reported lack of guidelines, organizational policies, separate recording, and registration, limited facilitated IPV training for providers, lack of referral network and specific referral systems or follow-up and delay in medico-legal reports for IPV victims were health system barriers that affected provider readiness in managing IPV [17, 21, 22].

A lack of provider readiness in managing IPV leads to ineffective responses to IPV disclosure or referrals for services which hinders victims from receiving the care and assistance they need [23]. Consequently, if the victims do not obtain proper treatment and management, they will continue suffering from physical, psychological, and sexual health problems [24]. Hence, to offer effective care and management for intimate partner victims in health facilities, it is necessary to assess health provider's readiness for proper intervention and management however, majority of studies about IPV conducted in Ethiopia focused on its prevalence, associated factors, and consequences, with only a few studies attempt to address the identification and treatment of IPV. To this end, this study intended to assess health providers' readiness in managing intimate partner violence in public health institutions at Hawassa, Ethiopia. In addition, we also examined factors associated with providers' perceived readiness to manage IPV

## Methods

### Study area and design

A cross-sectional study was conducted among health care providers working in selected public health institutions of Hawassa city administration from February 08 to March 08/2021. Hawassa city is located at 273 km from Addis Ababa, the capital city of Ethiopia. The city is served as the capital of southern nations, nationalities, and people region and Sidama region. The city administration is divided into eight sub-cities and 32 kebeles. According to the Hawassa City Administration health department 2021 population profile, the total population of Hawassa city was 394,057. There were 3 governmental hospitals and 10 health centers. A total of 596 general practitioners, health officers, Midwives, and Nurses) were currently working in the hospitals and health centers.

### Sample size and sampling technique

The sample size was calculated using the single population mean formula with the following assumptions; level of significance of 95%, sample variance (standard deviation) of 0.5 and 5% margin of error. Since there was no similar study in the study area, a pilot study was conducted among 10% of the source population to determine variance or standard deviation. Accordingly using a 10% non-response rate, the calculated sample size was found to be 424. The sample size was proportionally allocated for the randomly selected health facilities. General practitioners, nurses, midwives, and health officers who provided consent and available at the time of data collection were included in the study. Health care providers in the study group who had less than six-months of work experience were excluded.

### Measurement

Readiness to Manage Intimate Partner Violence Survey (PREMIS) tool [25]. This tool is developed and validated in the USA. However, validity was checked in Nigeria. Previously, the tool was used to assess health care providers such as (physicians, nurses, and midwives' preparedness to manage IPV in different countries including Ethiopia [15, 17, 26]. The tool assesses the level of health provider's readiness in managing IPV. The tool consists four subscales. The first subscale measures perceived readiness which is health care providers feeling regarding their readiness to manage IPV using nine items based on a 7-point Likert scale. Second, perceived knowledge subscale determines how much respondents feel as they know about IPV using 12 items with 7-point Likert scale. Third, the Attitude subscale was constructed from 13 items, reflecting belief and /or opinion of providers toward IPV management by 4-point Likert scale. Lastly, the practice subscale had 15 items with a 5-point Likert scale to measure current practice of providers in managing IPV. The higher scores indicated more readiness of providers. The tool showed evidence of high internal consistency and reliability with Cronbach's alpha > 0.65.

### Data collection

An anonymous self-administered questionnaire was used to collect data. Six BSc nurses were facilitating data collection process since it was self-administering and the entire data collection process was closely monitored by the principal investigators and two supervisors. Two days of training on the data collection were provided for the data collectors and supervisors. Pretest was conducted before actual data collection was conducted.

## Data analysis

The collected data were entered into Epi-data version 4.6.2 to look for outliers, missing values, and inconsistency, then exported into SPSS version 25.0 for analysis. Descriptive statistics were used to summarize the socio-demographic characteristics. Perceived knowledge, attitude, and practical section were summarized by figure and table. All linear regression assumptions were confirmed. The scatter plot was used to verify the linearity between the dependent and independent variables, histogram /Q-Q plot were used to check the multivariate normality. The scatter plot was used to assess for homoscedasticity (constant variance), which showed that the residuals were similar around the regression line. The Variance Inflation Factor (VIF) was used to determine multicollinearity, and it showed that there is no multicollinearity in the final model because the value for each variable was less than 5. A Scatter plot was used to check for the presence or absence of outliers, the finding indicated that all assumptions were fitted.

Simple linear regression was done to select candidate variables for multivariable linear regression. All variables having P-value ≤ 0.25 during the simple linear regression analysis and deemed important variable by the researcher were considered as candidates variables for the multivariable linear regression. After the multivariable linear regression analysis, variables having p-values <0.05 were considered as having a statistically significant association with the dependent variable. The strength of association between independent and dependent variables was assessed by using unstandardized β with 95% CI.

The Research Ethics Committee of Addis Ababa University, College of Health Science, department of Midwifery granted ethical clearance. An official letter of cooperation was received from the Health Department of Hawassa City Administration, and permissions for data collection were acquired from each health facility. Importance and objective of the study were explained to the study participant before data collection, and informed verbal consent was obtained. Participants were also informed that participation was voluntary and that they had the right to withdraw from the study at any time they wanted.

## Results

### Socio-demographic characteristics of the study participant

A total of 424 health care providers took part in this study. The majority of the respondents 238(56.1%) were females, more than half 221(52.1%) were protestant and 218 (51.4%) were married /living together. The mean age was 31.12 years+ 5.45, with the range of 20–51 and 204 (48.1%) of them were between the ages of 20 and 29 (Table 1).

### Perceived knowledge of the respondents in managing IPV

The mean score of Perceived provider's knowledge was 25.3± 6.6. About 203 (47.88%) of respondents had little knowledge about the legal reporting requirements for IPV, while 175 (41.3%) did not know the signs and symptoms of IPV. And 136 (32.07%) had a moderate amount of knowledge of referral sources for IPV victims. In turn, 191 people (45.05%) were very knowledgeable about the relationship between IPV and pregnancy. Finally, 207 (48.8%) of respondents had no idea about what their role was in detecting IPV (Fig 1).

### Attitudes of the respondents in managing IPV

The mean score of attitudes was 36.3 ± 6.12. About 362 (85.4%) of respondents agreed that they have to ask about intimate partner abuse (IPV). Three hundred twenty-four (76.4%) of the participants said they were not comfortable discussing the topic of IPV. The majority of respondents 311(73.3%) said they were unaware of the state's legal requirements for reporting

**Table 1. Socio-demographic characteristics of respondents in managing IPV in public health institutions of Hawassa, Ethiopia.** (n = 424).

| Variables | Category | Frequency | Percentage |
|---|---|---|---|
| Sex of respondent | Male | 186 | 43.9 |
| | Female | 238 | 56.1 |
| Age (completed years) | 20–29 | 204 | 48.1 |
| | 30–39 | 196 | 46.2 |
| | ≥ 40 | 24 | 5.7 |
| Religion | Protestant | 221 | 52.1 |
| | Orthodox | 153 | 36.1 |
| | Muslim | 28 | 6.6 |
| | Catholic | 19 | 4.5 |
| | Others | 3 | 0.7 |
| Marital Status of the respondent | Single | 196 | 46.2 |
| | Married/cohabited | 218 | 51.4 |
| | Divorced | 10 | 2.4 |
| Profession | General practitioners | 33 | 7.8 |
| | Health officers | 83 | 19.6 |
| | Midwives | 70 | 16.5 |
| | Nurses | 238 | 56.1 |
| Educational level | Master | 36 | 8.5 |
| | Bachelor | 366 | 86.3 |
| | Diploma | 22 | 5.2 |
| Work Experience | 1–5 years | 174 | 41.0 |
| | 6–10 years | 200 | 47.2 |
| | 11–15 years | 32 | 7.5 |
| | >15 | 18 | 4.5 |
| Current Practical area | Outpatient department | 143 | 33.7 |
| | Maternal and child health | 146 | 34.4 |
| | Emergency | 135 | 31.4 |
| Received IPV training | Yes | 64 | 15.1 |
| | No | 360 | 84.9 |

IPV. Only 162 (38.2%) of respondents accepted that their workplace allows them to respond to IPV (Table 2).

## The practice of respondents in IPV management

The mean score of practice was 48.7 + 14.3. According to this report, 150 (35.4%) of respondents were almost always asked about IPV when they saw a patient with injury, and 127 (30.0%) were almost always asked about IPV when they saw a patient with depression. The majority of respondents 130 (30.7%) were reported as they correctly registered patient statements. About 145 (34.2%) of respondents said they had never used a body map to record a patient's injury. Similarly, 185 (43.6%) of them never photographed the victim's injuries for inclusion in the document (Table 3).

## Health provider's perceived readiness in managing IPV

The mean score of perceived readiness in managing IPV was 26.18±6.69 and 95%CI (25.54, 26.8.2) with a range of 11–47. The mean score for appropriately responding to disclosures of

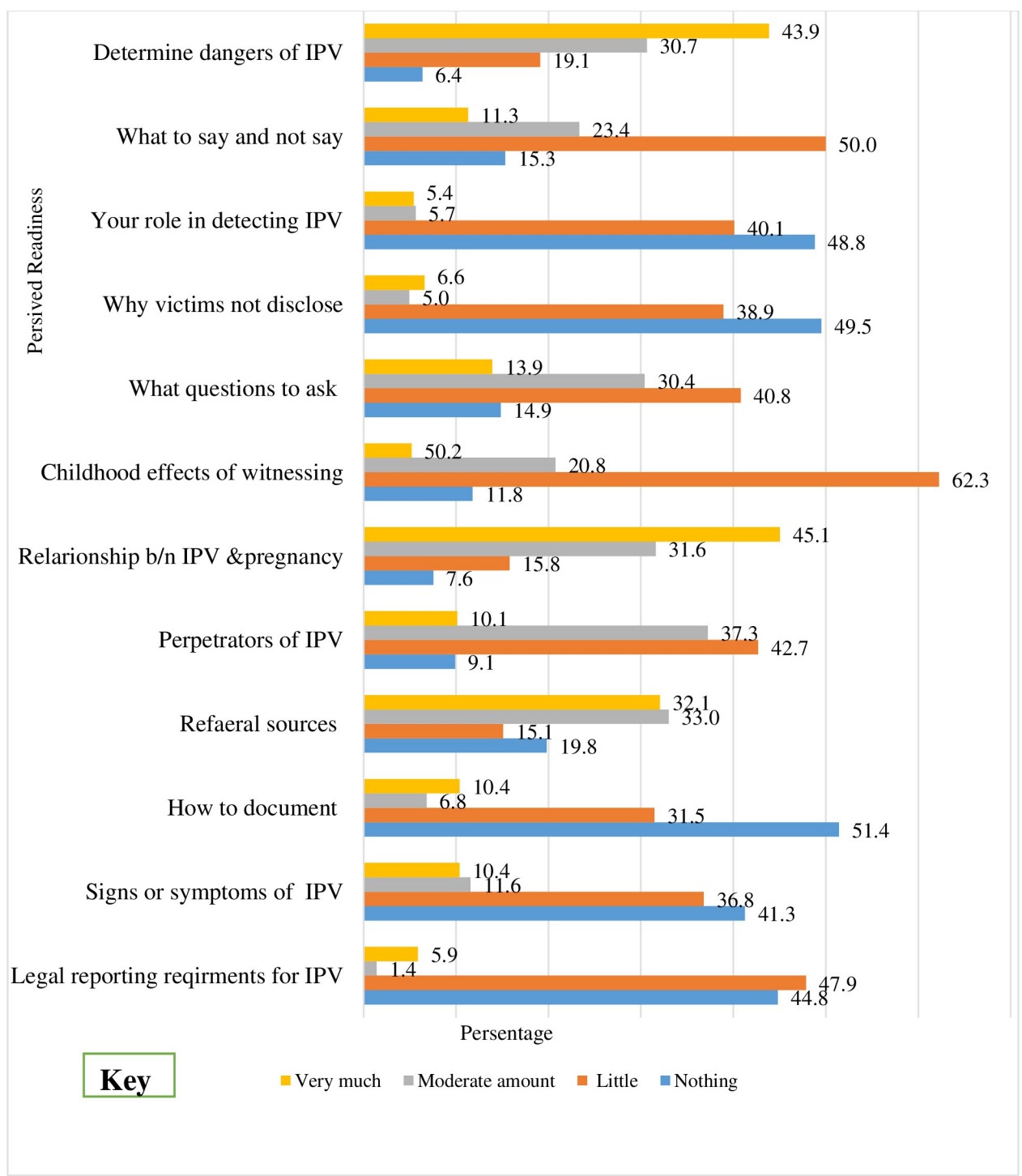

**Fig 1. Perceived knowledge of respondents for IPV management in public health institutions.** (n = 424).

**Table 2. Attitude of respondents towards managing IPV in public health institutions of Hawassa, Ethiopia (n = 424).**

| Variable | Disagree | | Agree | |
|---|---|---|---|---|
| | Frequency | % | Frequency | % |
| Health providers have a responsibility to ask about IPV | 62 | 14.6 | 362 | 85.4 |
| I feel comfortable discussing IPV | 324 | 76.4 | 100 | 23.6 |
| I am able to gather the necessary information to identify IPV | 310 | 73.1 | 114 | 26.9 |
| I can match therapeutic information to an IPV patient | 104 | 24.5 | 320 | 75.5 |
| I can make an appropriate referral to the service | 89 | 21.0 | 334 | 79.0 |
| IPV victims have the right to make their own decision | 100 | 23.6 | 324 | 76.4 |
| I am aware of the legal requirements for reporting IPV | 311 | 73.7 | 113 | 26.7 |
| My workplace encourages to respond to IPV | 262 | 61.8 | 162 | 38.2 |
| There is private space to provide care for IPV victims | 279 | 65.8 | 145 | 34.2 |
| Victims of IPV can leave the r/s if they want | 90 | 21.2 | 334 | 78.8 |
| Victims often have a valid reason for reaming in the violent relationship | 320 | 75.5 | 104 | 24.5 |
| If IPV patients refuse to discuss violence providers should treat only injury | 118 | 27.8 | 306 | 72.2 |
| If IPV victims remain in the r/s after repeated episodes, they mustn't accept responsibility for the violence | 289 | 68.2 | 135 | 31.8 |

**Table 3. The practice of respondents in managing IPV in public health institutions of Hawassa, Ethiopia (n = 424).**

| Variables | Never | Seldom | Some time | Nearly Always | Always |
|---|---|---|---|---|---|
| | No (%) | No (%) | No (%) | No (%) | No (%) |
| Asked about IPV when they see patients with injury | 27(6.4) | 17(4.0) | 105(24.8) | 150(35.4) | 125(29.5) |
| Patient with chronic pelvic pain | 55(13.0) | 35(8.3) | 104(24.5) | 135(31.8) | 95(22.4) |
| Patient with irritable bowel | 101(23.8) | 31(7.3) | 78(18.4) | 142(33.5) | 72(17.0) |
| Patient with headache | 54(12.7) | 17(4.0) | 145(34.2) | 100(23.6) | 108(25.5) |
| Patient with depression | 41(9.7) | 25(5.9) | 120(28.3) | 127(30.0) | 111(26.2) |
| Patient with an eating disorder | 120(28.3) | 40(9.4) | 126(29.7) | 90(21.2) | 48(11.3) |
| Documented patient statement | 130(24.3) | 36(8.5) | 39(9.2) | 116(27.4) | 130(30.7) |
| Used body map to document patient injuries | 145(34.2) | 36(8.5) | 77(18.5) | 99(23.3) | 67(15.8) |
| Photographed victim's injuries to include in the chart | 185(43.6) | 46(10.8) | 49(11.6) | 77(18.2) | 67(15.8) |
| Notified appropriate authorities when mandated | 102(24.1) | 50(11.8) | 64(15.1) | 94(22.2) | 114(26.90) |
| Conducted a safety assessment for the victim | 97(22.9) | 36(8.5) | 89(21.0) | 111(26.2) | 91(21.5) |
| Helped an IPV victim to develop a safety plan | 73(17.2) | 41(9.7) | 74(17.5) | 146(34.4) | 90(21.2) |
| Offered validating statements | 67(15.8) | 26(6.1) | 107(25.2) | 127(30.0) | 97(22.9) |
| Provided basic information about IPV | 68(16.0) | 20(4.7) | 101(23.8) | 126(29.7) | 107(25.7) |
| Provided referral and/or resource information | 63(14.9) | 27(6.4) | 66(15.6) | 147(34.7) | 121(28.5) |

IPV victims were 3.26. Correspondingly, the Mean score for identifying IPV indicators based on patient history and physical examination was 3.37. The list means the score was measured for item stetting fulfill state reporting requirements for IPV was 2.17. The total mean score for each item was 2.94 (Fig 2).

## Health providers readiness by socio demographic characters

In order to compare health care providers by socio-demographic characters we classified the level of readiness based on mean score of perceived readiness. Accordingly, health providers scored above 26.2 mean value were considered to be ready to manage IPV. About (57.7%) females were more ready to manage IPV than males. Provider who has Married/cohabited

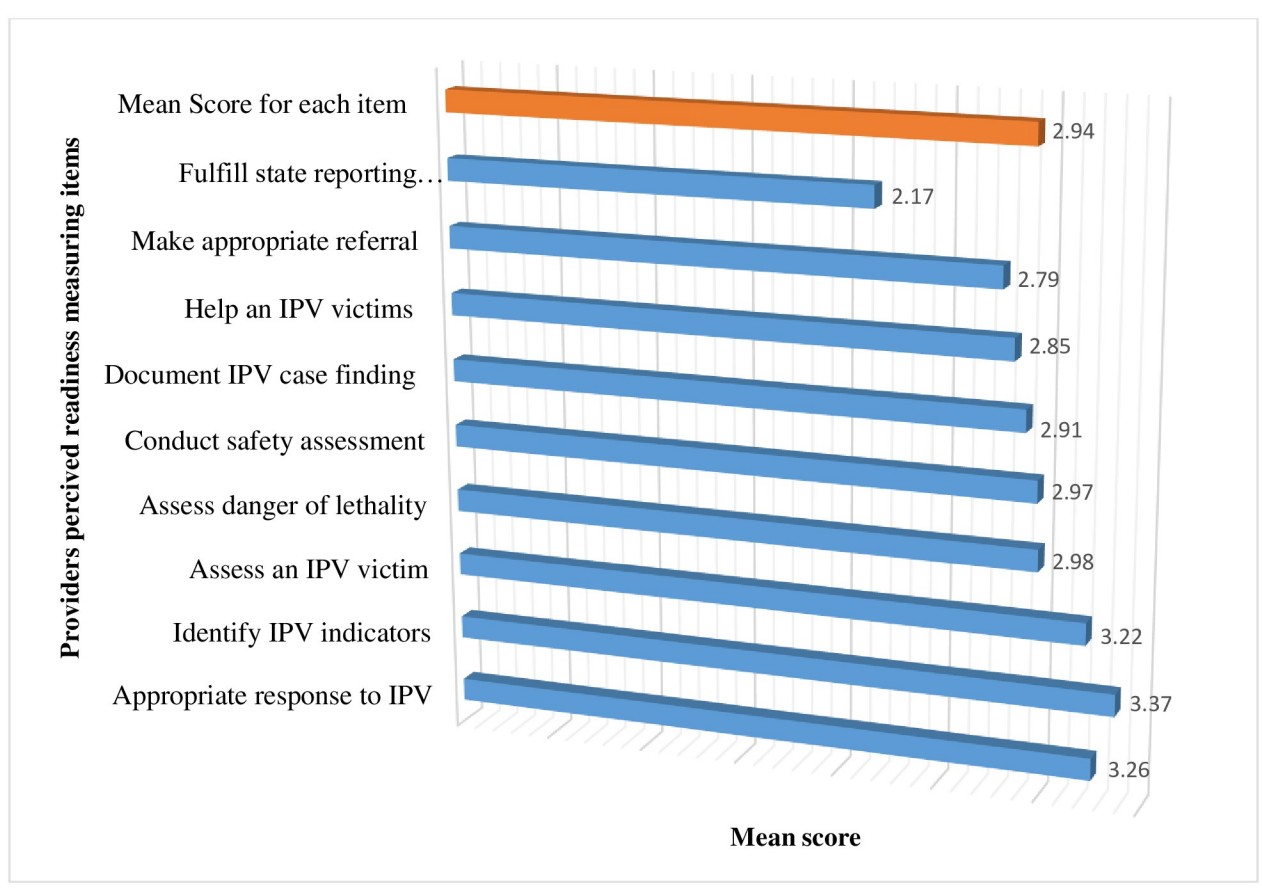

**Fig 2. Perceived readiness of respondents in managing IPV in health institutions.** (n = 424).

(52.6%) marital status more ready to manage IPV than single and divorced. Midwifes (60.2%) were more ready to manage IPV than general practitioners, health officers, and nurses. The level of readiness based of place of work during data collection was almost similar. Table 4

## Factors associated with provider's perceived readiness in managing IPV

Simple linear regression analysis was done to find the association of socio-demographic characteristics, health providers related and health facilities related factors on provider's perceived readiness in managing IPV. During simple linear regression availability of protocol at practical area, receiving IPV training, perceived knowledge of providers in managing IPV and attitude towards IPV management showed statistically significant association with provider's perceived readiness in managing IPV.

Eleven variables entered into the final multiple linear regression model. The final model has fulfilled all assumptions of normality and all the independent variables scored VIF less than five indicating non-existence of multicollinearity. Adjusted R square revealed that 68.0% of the dependent variable could be explained by the independent variable. The overall regression model was good fit with F value of 3.48 and p value < 0.001. During the final model sex, marital status, profession, clinical work experience, practice and familiarity with institution's policies regarding management of IPV were not found to be significant predictor.

**Table 4. Level of health providers readiness by sociodemographic characteristics of health providers working at public health institutions Hawassa, Ethiopia.**

| Variables | categories | Ready to manage IPV | |
|---|---|---|---|
| | | No | % |
| Sex | Male | 83 | 42.3 |
| | Female | 113 | 57.7 |
| Age | 20–29 | 100 | 51.0 |
| | 30–39 | 84 | 42.9 |
| | ≥ 40 | 12 | 6.1 |
| Marital status | Single | 89 | 45.4 |
| | Married/cohabited | 103 | 52.6 |
| | Divorced | 4 | 2.0 |
| Profession | General practitioner | 16 | 8.2 |
| | Health officers | 31 | 15.8 |
| | Midwives | 118 | 60.2 |
| | Nurses | 31 | 15.8 |
| Educational level | Master | 35 | 17.9 |
| | Bachelor | 151 | 77.0 |
| | Diploma | 10 | 5.1 |
| Working experience | 1–5 years | 81 | 41.3 |
| | 6–10 years | 91 | 46.4 |
| | 11–15 years | 16 | 8.2 |
| | >15 | 8 | 4.1 |
| Current working area | Outpatient department | 65 | 33.2 |
| | Maternal and child health | 68 | 34.7 |
| | Emergency | 63 | 32.1 |
| Receiving IPV training | Yes | 167 | 85.2 |
| | No | 29 | 14.8 |

According to this particular study, the perceived readiness score of health providers in managing IPV increased by 0.18 for every ten years of age (P = 0.017: 95% CI: 0.03, 0.32). Correspondingly, Providers who feel as they know about IPV management (perceived knowledge) had an increased perceived readiness score by 0.29 (P 0.001: 95% CI: 0.20, 0.38). However, the perceived readiness score of providers who did not receive IPV training fell by 2.10 (P = 0.013: 95% CI: -3.75, -0.45). Also, the Absence of protocol for dealing with IPV management in the institutions decreases the provider's perceived readiness score by 1.36 (P = 0.049: 95% CI of β: -2.71, -0.01). Similarly, poor provider attitude decreases perceived readiness score by 0.11 (P = 0.027: 95%CI: 0.01, 0.21) (Table 5).

## Discussion

This study attempts to assess health providers' perceived readiness in managing IPV in public health institutions of Ethiopian city. Perceived readiness in managing IPV signifies the provider's beliefs on how prepared to identify the case, provide first-line support, and deliver medical care including referral and follow-up. Indeed, readiness can be thought of as a motivator for people to make positive changes [12]. Therefore, assessing a provider's readiness in managing IPV can be an effective first step in ensuring an adequate response to IPV victims at the health facility level.

**Table 5. Multiple linear regression analysis for factors in managing IPV in public health institutions of Hawassa, Ethiopia (n = 424).**

| Variable | Category | Unstandardized Coefficients ß | P-value | 95%CI of β |
|---|---|---|---|---|
| Sex | Male | 1 | | |
| | Female | 0.06 | 0.917 | (-1.14, 1.27) |
| Age | | 0.18 | **0.017** | (0.03, 0.32) |
| Marital status | Single | 1 | | |
| | Married/cohabited | -0.53 | 0.428 | (-1.83, 0.78) |
| | Divorce | -0.67 | 0.669 | (-4.95, 3.18) |
| Profession | General practitioner | -1.30 | 0.274 | (-3.62, 1.03) |
| | Health officer | -0.84 | 0.309 | (-2.46, 0.78) |
| | Midwives | 0.41 | 0.634 | (-1.27, 2.09) |
| Clinical work experience | 1–5 | 1 | | |
| | 6–10 | -0.70 | 0.356 | (-2.17, 0.78) |
| | 11–15 | 0.01 | 0.997 | (-2.54, 2.55) |
| | >15 | -1.98 | 0.325 | (-5. 94, 1.97) |
| Received IPV training | Yes | 1 | | |
| | No | -2.10 | **0.013** | (-3.746, -0.45) |
| Availability of protocol for dealing with IPV | Yes | 1 | | |
| | No | -1.36 | **0.049** | (-2.71, -0.01) |
| Familiarity with institutional policies | Yes | 0.711 | 0.362 | (-0.82, 2.24) |
| | No | 1 | | |
| Perceived knowledge | | 0.29 | **< 0.001** | (0.20, 0.38) |
| Attitude | | 0.11 | **0.027** | (0.01, 0.21) |
| Practice | | 0.08 | 0.061 | (0.05, 0.23) |

The total mean score for each item was 2.94 which was low as compared to previous studies in Saudi Arabia and the United States, which scored 3.10 and 4.98 respectively [11, 23]. This may be due to a disparity in healthcare attention to violence. This indicates the need for promoting an IPV screening program.

According to the current results, the mean perceived readiness score in managing IPV increases as providers' ages increase. This may be related to a change in attitude and work experiences. These can be explained by the fact that when a provider's maturity and years of experience raise, they gain a better understanding of IPV and more positive attitudes toward it. This finding was in line with the findings of a Nigerian report [27]. Similarly, a Tanzanian study found that providers with longer work experience were more likely to have an opportunity for IPV in-service training [15]. This may be because early on, providers lacked some basic knowledge about IPV management, and as they get older, this knowledge gap can close with experience and training.

The findings of this study revealed the association between training and perceived readiness in managing IPV (P-value = 0.013). Not receiving in-service IPV training reduced providers' perceived readiness score. This finding is consistent with previous studies that have linked training to improved IPV identification in a health setting. The training was associated with a greater perceived preparation score and more active identification of IPV (p <0.001) [23]. This is concurrent with A study conducted in Australia [27], Tanzania [15], and Ethiopia [26]. This may be due to providers receiving little or no classroom training on how to identify, manage, and refer patients who are being abused by an intimate partner.

Furthermore, this study's findings showed that the absence of protocol dealing with IPV management in facilities condensed perceived providers' readiness in managing IPV. This

finding is supported by a study conducted in the US (P = 0.001) and Systematic Review [23, 28]. This may be due to Standardized protocols being important to guide service delivery and support the delivery of safe, good quality, respectful and effective health care that is consistent across locations.

The findings indicated providers' perceived knowledge was associated with their perceived readiness (P<0.001). Which suggested perceived readiness of the providers regarding the management of IPV victims was significantly predicted by the level of their perceived knowledge of the issue. This finding was in line with a study done in Sweden having obtained knowledge by themselves to increase providers' perceived readiness score in managing IPV (p <0.001) [29]. Correspondingly, the Nigerian study supports the finding, that perceived preparedness for IPV management is significantly affected by the level of perceived knowledge (P<0.001) [17]. This may be due to perceived knowledge being the initial prerequisite requirement for perceived readiness in managing IPV.

Provider's attitude was significantly associated with their perceived readiness in managing IPV (P 0.027). In this study, providers did not feel comfortable discussing IPV. This may be due to either lack of knowledge about IPV or fear of legal aspects of the issue. This highlights that the reason why providers are not dealing with IPV is they have low confidence and encouragement, and the way they perceive readiness in managing IPV could be affected. This finding was in line with previous studies [11, 30]. This might be due to awareness of one's attitudes playing an important role in one's perceptions.

## Strength of the study

The findings of this study open the way for further analytical studies to identify determinants of provider readiness. The study incorporates different professionals: general practitioners, health officers, nurses, and midwives which can help to make an appropriate inference.

## Limitations of the study

Hard to establish a cause-and-effect relationship since it is a cross-sectional study design.

## Conclusion

Intimate partner violence (IPV) is a public health problem that disproportionately affects women's health and well-being, and the effects of IPV have been well reported throughout the literature. The findings of this study revealed that providers had limited perceived readiness to manage IPV. Provider's age, lack of IPV related training, absence of protocol in facilities, perceived knowledge, and attitude of health provider towards IPV management were identified to be independent predictors of perceived readiness in managing IPV.

In order to improve readiness of health care providers for managing IPV, health planer and policy makers needs to develop and implement formal, written protocol detailing the specific procedures for identifying and managing IPV cases, facilitate provision of training to improve health providers readiness in managing IPV, and familiarize health providers with institution's policies and encourage for successful implementation of IPV protocol. In addition, national study using mixed method is needed to examine provider's readiness in managing IPV in the country and explore factors not addressed in the current study. reasons.

## Supporting information

**S1 File. Assumptions and the model fitness t-test and plot.**
(DOCX)

**S1 Checklist. STROBE statement—Checklist of items that should be included in reports of observational studies.**
(DOCX)

**S1 Data.**
(ZIP)

## Acknowledgments

We are thankful to the staff of Adare Hospital and Health centers which were included in this study and found under the Hawassa city administration health department for their valuable contributions. Finally, the authors are also thankful to the supervisors and data collectors.

## Author Contributions

**Conceptualization:** Lidiya Teshome, Haweni Adugna, Leul Deribe.

**Data curation:** Lidiya Teshome, Haweni Adugna, Leul Deribe.

**Formal analysis:** Lidiya Teshome, Haweni Adugna, Leul Deribe.

**Funding acquisition:** Lidiya Teshome, Haweni Adugna, Leul Deribe.

**Investigation:** Lidiya Teshome, Haweni Adugna, Leul Deribe.

**Methodology:** Lidiya Teshome, Haweni Adugna, Leul Deribe.

**Project administration:** Lidiya Teshome, Haweni Adugna, Leul Deribe.

**Resources:** Lidiya Teshome, Haweni Adugna, Leul Deribe.

**Software:** Lidiya Teshome, Haweni Adugna, Leul Deribe.

**Supervision:** Haweni Adugna, Leul Deribe.

**Validation:** Lidiya Teshome, Haweni Adugna, Leul Deribe.

**Visualization:** Lidiya Teshome, Haweni Adugna, Leul Deribe.

**Writing – original draft:** Lidiya Teshome, Haweni Adugna, Leul Deribe.

**Writing – review & editing:** Lidiya Teshome, Haweni Adugna, Leul Deribe.

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
