## [Decision Letter · Decision Letter 0]

13 Dec 2022

PONE-D-22-15821Health Provider’s Readiness in Managing Intimate Partner Violence in Public Health Institutions at Hawassa City, Sidama Region, Ethiopia, 2021.PLOS ONE

Dear Dr. Deribe,

Thank you for submitting your manuscript to PLOS ONE. After careful consideration, we feel that it has merit but does not fully meet PLOS ONE’s publication criteria as it currently stands. Therefore, we invite you to submit a revised version of the manuscript that addresses the points raised during the review process.

We look forward to receiving your revised manuscript.

Kind regards,

Humayun Kabir, MSc in Epidemiology

Academic Editor

PLOS ONE

Journal Requirements:

2 Your ethics statement should only appear in the Methods section of your manuscript. If your ethics statement is written in any section besides the Methods, please delete it from any other section.

Reviewers' comments:

Reviewer's Responses to Questions

**Comments to the Author**

1. Is the manuscript technically sound, and do the data support the conclusions?

Reviewer #1: Yes

Reviewer #2: Yes

2. Has the statistical analysis been performed appropriately and rigorously? 

Reviewer #1: Yes

Reviewer #2: Yes

3. Have the authors made all data underlying the findings in their manuscript fully available?

Reviewer #1: No

Reviewer #2: Yes

4. Is the manuscript presented in an intelligible fashion and written in standard English?

Reviewer #1: No

Reviewer #2: No

5. Review Comments to the Author

Reviewer #1: Intimate partner violence is a serious public health issue and provider knowledge and prepardeness to care for patients experiencing IPV is an important part of the public health response to this issue. This manuscript presents data regarding the readiness of providers in Hawassa City, Ethiopia. The manuscript could be improved by addressing the following issues:

The sample size calculation is presented before any information about what data is being collected. Mean of what?? The manuscript should be reorganized so that the sample size calculation comes after the variables.

Study variables: The PREMIS instrument is identified after the description of the study variables. The use of a standardized instrument is a strength of this study. The instrument identification and description should be at the start of the variable section.

Strength of Study and Limitations should not be bulletted lists.

Overall, the manuscript needs to be edited for English language. There are sections that are hard to read and sentences that don't make sense.

Reviewer #2: This well-conducted research examines health provider’s readiness to address IPV in Ethiopia. This topic is clearly of relevance to PLOS-One.

Formatting: OK.

Writing: Some grammatical errors (i.e. needed apostrophes), awkward or stilted language and odd phrasing (i.e. “needed to upsurge” on page 1 in the abstract. This submission need to be edited for English syntax.

Too many acronyms throughout (i.e. OPD & MCH). Better to write out names except for common acronyms such as IPV.

Title: Too complex. Perhaps just note Ethiopia (rather than city and state details)

and leave off the year.

Abstract: Clear and detailed. However, it is unusual to include such details as the p-value for significance and confidence interviews. Keep such details to the body of the submission.

Introduction: Some strongly worded statements that are questionable. Looking at only the first paragraph, stating that IPV is the “most prevalent form of violence” is not usually stated as such. Also “It occurs without exclusion”… is an unusual way to make the point that it is common. “Almost all violent behavior in IPV psychological violence” – is not stated clearly. What does this mean? Globally, should be followed by a comma.

In the second paragraph, the point about homicides is appropriate but the subsequent connection to medical treatment and support does not follow.

In the third paragraph, “Health providers are front lines responsible” … This does not make sense.

Better to first introduce the global statistics and issues related to IPV. Then add the Ethiopian context in a separate section.

Research Methods: It is unusual to include a statistical formula regarding sample size. Simply delete and describe in the narrative.

The PREMIS tool is a strong choice. Was the full scale used or not? It is confusing to present the study variables separately if they are directly derived from the PREMIS (Which seems the case). Delete the section on study variables. Consolidate this information.

Results: Great sample size (N = 428). The data analysis is largely simply descriptive. This is OK but could be more interesting i.e. if there was a logical comparison such as by gender or health care practice area.

Discussion: The comparison of the results to other PREMIS studies, especially other African studies, is interesting and appropriate.

Summary: In summary, this Ethiopian study of IPV readiness with a large sample size using a well-respected tool, the PREMIS, has much to recommend it. However, as currently written it needs a thorough edit for English syntax.

References: Correctly formatted although digital object identifiers (dois) could be used as well.

Specific Issues: (no page numbers or line numbers so difficult to pinpoint specific issues.

p. 1 Abstract Introduction. The phrase “against all women” is incorrect. IPV is a significant problem but not for all.

6. PLOS authors have the option to publish the peer review history of their article (what does this mean?). If published, this will include your full peer review and any attached files.

Reviewer #1: No

Reviewer #2: No

---

## [Author Response · Author response to Decision Letter 0]

14 Mar 2023

Response to reviewers’ comment 

We thank both reviewers for the interest they shown to review our paper and for their valuable comments. We made all modification based on the comment provided and the change made indicated in in line with each comment. 

Review Comments to the Author

Reviewer #1: Intimate partner violence is a serious public health issue and provider knowledge and preparedness to care for patients experiencing IPV is an important part of the public health response to this issue. This manuscript presents data regarding the readiness of providers in Hawassa City, Ethiopia. The manuscript could be improved by addressing the following issues:

The sample size calculation is presented before any information about what data is being collected. Mean of what?? The manuscript should be reorganized so that the sample size calculation comes after the variables.

1. We used a single population mean formula to calculate our sample size. However, there was no similar study done to determine the variance. Then a pilot study was conducted to determine mean and variance of perceived readiness in managing IPV. on 10% of the source population. 

Study variables: The PREMIS instrument is identified after the description of the study variables. The use of a standardized instrument is a strength of this study. The instrument identification and description should be at the start of the variable section.

Strength of Study and Limitations should not be bulletted lists.

2. We avoid the bullets 

Overall, the manuscript needs to be edited for English language. There are sections that are hard to read and sentences that don't make sense.

3. We revised the language 

Reviewer #2: This well-conducted research examines health provider’s readiness to address IPV in Ethiopia. This topic is clearly of relevance to PLOS-One.Formatting: OK.

Writing: Some grammatical errors (i.e. needed apostrophes), awkward or stilted language and odd phrasing (i.e. “needed to upsurge” on page 1 in the abstract. This submission need to be edited for English syntax.

Too many acronyms throughout (i.e. OPD & MCH). Better to write out names except for common acronyms such as IPV.

4. Uncommon acronyms removed

Title: Too complex. Perhaps just note Ethiopia (rather than city and state details)

and leave off the year.

5. The title modified to 

Health providers readiness in managing intimate partner violence in public health institutions, Ethiopia.

Abstract: Clear and detailed. However, it is unusual to include such details as the p-value for significance and confidence interviews. Keep such details to the body of the submission.

6. p-value and confidence interval is removed 

Introduction: Some strongly worded statements that are questionable. Looking at only the first paragraph, stating that IPV is the “most prevalent form of violence” is not usually stated as such. Also “It occurs without exclusion”… is an unusual way to make the point that it is common. “Almost all violent behavior in IPV psychological violence” – is not stated clearly. What does this mean? Globally, should be followed by a comma.

7. Revised 

In the second paragraph, the point about homicides is appropriate but the subsequent connection to medical treatment and support does not follow.

8. Revised 

In the third paragraph, “Health providers are front lines responsible” … This does not make sense.

9. Revised 

Better to first introduce the global statistics and issues related to IPV. Then add the Ethiopian context in a separate section.

10. Revised 

Research Methods: It is unusual to include a statistical formula regarding sample size. Simply delete and describe in the narrative.

11. The formula for sample size removed 

The PREMIS tool is a strong choice. Was the full scale used or not? It is confusing to present the study variables separately if they are directly derived from the PREMIS (Which seems the case). Delete the section on study variables. Consolidate this information.

12. The tool has four sections 

• perceived readiness (9 items)

• Perceived knowledge (12 items)

• Attitude section (13 items)

• Practice section (11 items)

The dependent variable in section is perceived readiness and we believed it is useful to descriptively present the other section of the tool 

Results: Great sample size (N = 428). The data analysis is largely simply descriptive. This is OK but could be more interesting i.e. if there was a logical comparison such as by gender or health care practice area.

Discussion: The comparison of the results to other PREMIS studies, especially other African studies, is interesting and appropriate.

Summary: In summary, this Ethiopian study of IPV readiness with a large sample size using a well-respected tool, the PREMIS, has much to recommend it. However, as currently written it needs a thorough edit for English syntax.

13. Recommendation added 

References: Correctly formatted although digital object identifiers (dois) could be used as well.

Specific Issues: (no page numbers or line numbers so difficult to pinpoint specific issues.

14. Reference revised and DOI number included 

p. 1 Abstract Introduction. The phrase “against all women” is incorrect. IPV is a significant problem but not for all.

15. Modified 

6. PLOS authors have the option to publish the peer review history of their article (what does this mean?). If published, this will include your full peer review and any attached files.

---

## [Decision Letter · Decision Letter 1]

15 May 2023

PONE-D-22-15821R1Health providers readiness in managing intimate partner violence in public health institutions, Ethiopia.PLOS ONE

Dear Dr. Deribe,

Thank you for submitting your manuscript to PLOS ONE. After careful consideration, we feel that it has merit but does not fully meet PLOS ONE’s publication criteria as it currently stands. Therefore, we invite you to submit a revised version of the manuscript that addresses the points raised during the review process.

We look forward to receiving your revised manuscript.

Kind regards,

Humayun Kabir

Academic Editor

PLOS ONE

Additional Editor Comments:

As the author did the linear regression, did they check the normality assumption? If yes, provide how they did; histogram/density or other plots are suggested as a supplementary file to see the outcome's distribution. 

Report all the other assumptions assessed before fitting the model, such as correlation, interaction, and confounding. If the correlation, interaction, and confounding were found, how was the variable selected in the model? 

After fitting the model, did the author check the fitness of the model?

Report and interpret the intercept, R Squared, and F value of the model. 

Reviewers' comments:

Reviewer's Responses to Questions

**Comments to the Author**

1. If the authors have adequately addressed your comments raised in a previous round of review and you feel that this manuscript is now acceptable for publication, you may indicate that here to bypass the “Comments to the Author” section, enter your conflict of interest statement in the “Confidential to Editor” section, and submit your "Accept" recommendation.

Reviewer #1: (No Response)

Reviewer #2: (No Response)

2. Is the manuscript technically sound, and do the data support the conclusions?

Reviewer #1: Partly

Reviewer #2: Partly

3. Has the statistical analysis been performed appropriately and rigorously? 

Reviewer #1: No

Reviewer #2: No

4. Have the authors made all data underlying the findings in their manuscript fully available?

Reviewer #1: (No Response)

Reviewer #2: Yes

5. Is the manuscript presented in an intelligible fashion and written in standard English?

Reviewer #1: No

Reviewer #2: No

6. Review Comments to the Author

Reviewer #1: The authors have addressed some of the reviewer comments. The following issues remain:

Abstract: "The strength of association...." The last two sentences have information that are not typically in an abstract. They should be removed.

Introduction: Ethiopia should be added to very end of the aim to make clear for all readers where this is.

Sample size calculation: Did the sample size calculation account for clustering? Given that the sample size was stratified, clustering should be included in the calculation.

Study Variables: The authors did not address the guidance from the reviewers regarding the structure of this section. The PREMIS instrument should be described before it's subscales are included.

Data collection: The data collection section is not about data collection. It is about the instrument. The content of this section should be moved up above the text in the study variable section. The use of the PREMIS is a strength of this study.

Methods: There is no data collection information. This should be added.

Reviewer #2: This interesting research examines health provider’s readiness to address IPV in Ethiopia. This topic is clearly of relevance to PLOS-One. The authors have responded conscientiously to a number of the reviewers’ comments and suggestions; but not all. The authors did not respond to at all to several of my suggestions.

The resubmission was confusing. The actual revised document was found after the first submission was presented (or perhaps this was the revised document without track changes… It was not immediately clear what was the case. Perhaps this was a function of the journal submission process, but it did add extra time to conduct the review. Further, the revised submission did not have lines on the side, so difficult to make specific suggestions.

Formatting: OK.

Writing: Still some English syntax problems (i.e., in the Abstract the sentence “Variable with the P- value of < 0.05 was considered a statistically…” needs editing i.e., “Variables with a p-value of <0.05 were…”. Also, as this is a commonly accepted p-value, no need to include this statement, especially not in an abstract.

The authors add a section to the end of the paper listing the acronyms used. However, this does not help the reader who must interpret a number of unusual acronyms throughout (i.e., OPD & MCH). As before, better to write out names except for common acronyms such as IPV.

Title: Better.

Abstract: Better

Introduction: Better.

Research Methods: As before, the PREMIS tool is a strong choice. It is confusing to present the study variables separately and first, if they are directly derived from the PREMIS (which seems the case). Present the PREMIS information first – then describe the subscales (which are the study variables) Consolidate this information.

Results: As before, this is a great sample size (N = 428). As before, the data analysis is largely simply descriptive. This is OK but could be more interesting i.e., if there was a logical comparison such as by gender or health care practice area. The authors have stuck with their original descriptive description of the demographics.

Discussion: The comparison of the results to other PREMIS studies, especially other African studies, is interesting and appropriate.

Summary: In summary and, as before, this Ethiopian study of IPV readiness with a large sample size using a well-respected tool, the PREMIS, has much to recommend it. However, the authors have not commented or revised several key recommendations from this reviewer such as reorganizing Methods to more directly credit the PREMIS tool and compare PREMIS results based on a key demographic variable such as gender or health care practice area. These would make the study less basic and potentially of more interest.

References: Correctly formatted.

7. PLOS authors have the option to publish the peer review history of their article (what does this mean?). If published, this will include your full peer review and any attached files.

Reviewer #1: No

Reviewer #2: No

<quillbot-extension-portal></quillbot-extension-portal>

---

## [Author Response · Author response to Decision Letter 1]

20 May 2023

Reviewer #1: The authors have addressed some of the reviewer comments. The following issues remain:

Abstract: "The strength of association...." The last two sentences have information that are not typically in an abstract. They should be removed.

the last two parts of abstract part removed

Introduction: Ethiopia should be added to very end of the aim to make clear for all readers where this is.

Ethiopia is added 

Sample size calculation: Did the sample size calculation account for clustering? Given that the sample size was stratified, clustering should be included in the calculation.

Sample size was calculated by using single population mean. After selecting health facilities by simple random sampling, the final sample is proportionally allocated for each facility. We did not use any stratification or clustering 

Study Variables: The authors did not address the guidance from the reviewers regarding the structure of this section. The PREMIS instrument should be described before it's subscales are included.

point is added about tool

Data collection: The data collection section is not about data collection. It is about the instrument. The content of this section should be moved up above the text in the study variable section. The use of the PREMIS is a strength of this study.

the content moved 

Information about PREMIS presented under heading measurement 

Methods: There is no data collection information. This should be added.

Added 

Reviewer #2: 

This interesting research examines health provider’s readiness to address IPV in Ethiopia. This topic is clearly of relevance to PLOS-One. The authors have responded conscientiously to a number of the reviewers’ comments and suggestions; but not all. The authors did not respond to at all to several of my suggestions.

The resubmission was confusing. The actual revised document was found after the first submission was presented (or perhaps this was the revised document without track changes… It was not immediately clear what was the case. Perhaps this was a function of the journal submission process, but it did add extra time to conduct the review. 

Further, the revised submission did not have lines on the side, so difficult to make specific suggestions.

Formatting: OK.

Writing: Still some English syntax problems (i.e., in the Abstract the sentence “Variable with the P- value of < 0.05 was considered a statistically…” needs editing i.e., “Variables with a p-value of <0.05 were…”. Also, as this is a commonly accepted p-value, no need to include this statement, especially not in an abstract.

removed and corrections made 

The authors add a section to the end of the paper listing the acronyms used. However, this does not help the reader who must interpret a number of unusual acronyms throughout (i.e., OPD & MCH). As before, better to write out names except for common acronyms such as IPV.

OPD & MCH removed 

Title: Better.

Abstract: Better

Introduction: Better.

Research Methods: As before, the PREMIS tool is a strong choice. It is confusing to present the study variables separately and first, if they are directly derived from the PREMIS (which seems the case). Present the PREMIS information first – then describe the subscales (which are the study variables) Consolidate this information.

Change made according to recommendation 

Results: As before, this is a great sample size (N = 428). As before, the data analysis is largely simply descriptive. This is OK but could be more interesting i.e., if there was a logical comparison such as by gender or health care practice area. The authors have stuck with their original descriptive description of the demographics.

We included a section with the title Health providers readiness by socio demographic characters and added a table to indicate comparison by sociodemographic characteristics 

Discussion: The comparison of the results to other PREMIS studies, especially other African studies, is interesting and appropriate.

Summary: In summary and, as before, this Ethiopian study of IPV readiness with a large sample size using a well-respected tool, the PREMIS, has much to recommend it. However, the authors have not commented or revised several key recommendations from this reviewer such as reorganizing Methods to more directly credit the PREMIS tool and compare PREMIS results based on a key demographic variable such as gender or health care practice area. These would make the study less basic and potentially of more interest.

All recommendation given are included 

References: Correctly formatted.

---

## [Editor Report · Decision Letter 2]

23 May 2023

PONE-D-22-15821R2Health providers readiness in managing intimate partner violence in public health institutions, Ethiopia.PLOS ONE

Dear Dr. Deribe,

Thank you for submitting your manuscript to PLOS ONE. After careful consideration, we feel that it has merit but does not fully meet PLOS ONE’s publication criteria as it currently stands. Therefore, we invite you to submit a revised version of the manuscript that addresses the points raised during the review process.

We look forward to receiving your revised manuscript.

Kind regards,

Humayun Kabir

Academic Editor

PLOS ONE

Additional Editor Comments:

As the author did the linear regression, did they check the normality and all other assumption of the model? If yes, provide how they did; histogram/density or other plots are suggested as a supplementary file to see the outcome's distribution.

Report all the other assumptions assessed before fitting the model, such as correlation, interaction, and confounding.

If the correlation, interaction, and confounding were found, how was the variable selected in the model?

After fitting the model, did the author check the fitness of the model?

Report and interpret the intercept, R Squared, and F value of the model.

Reporting guidelines STOBE is requested to use.

<quillbot-extension-portal></quillbot-extension-portal>

---

## [Author Response · Author response to Decision Letter 2]

15 Aug 2023

Response 

All feedbacks incorporated in the method and result section. The following statement added in the method, data analysis section 

All variables having P-value ≤ 0.25 during the simple linear regression analysis and deemed important variable by the researcher were considered as candidates variables for the multivariable linear regression.

And the following two paragraphs added in the Factors associated with provider’s perceived readiness in managing IPV

Simple linear regression analysis was done to find the association of socio-demographic characteristics, health providers related and health facilities related factors on provider’s perceived readiness in managing IPV. During simple linear regression availability of protocol at practical area, receiving IPV training, perceived knowledge of providers in managing IPV and attitude towards IPV management showed statistically significant association with provider’s perceived readiness in managing IPV. 

Eleven variables entered into the final multiple linear regression model. The final model has fulfilled all assumptions of normality and all the independent variables scored VIF less than five indicating non‐existence of multicollinearity. The model can explain 70% of the variation in provider’s perceived readiness. The overall regression model was good fit with F value of 3.48 and p value < 0.001. During the final model sex, marital status, profession, clinical work experience, practice and familiarity with institution’s policies regarding management of IPV were not found to be significant predictor.

---

## [Editor Report · Decision Letter 3]

21 Aug 2023

PONE-D-22-15821R3Health providers readiness in managing intimate partner violence in public health institutions, Ethiopia.PLOS ONE

Dear Dr. Deribe,

Thank you for submitting your manuscript to PLOS ONE. After careful consideration, we feel that it has merit but does not fully meet PLOS ONE’s publication criteria as it currently stands. Therefore, we invite you to submit a revised version of the manuscript that addresses the points raised during the review process.

ACADEMIC EDITOR:"The model can explain 70% of the variation in provider’s

perceived readiness" Please report the R-squared value in regard to 70% of the explained variance. 

all the model assumptions and the model fitness t-test and plot should be provided as supplementary files for transparency. 

Submit the STOBE checklist as a supplementary file. 

We look forward to receiving your revised manuscript.

Kind regards,

Humayun Kabir

Academic Editor

PLOS ONE
---

## [Author Response · Author response to Decision Letter 3]

6 Oct 2023

Response to ACADEMIC EDITOR:

"The model can explain 70% of the variation in provider’s

perceived readiness" Please report the R-squared value in regard to 70% of the explained variance. 

Response: 

- Adjusted R square revealed that 68.0% of the dependent variable could be explained by the independent variable. 

all the model assumptions and the model fitness t-test and plot should be provided as supplementary files for transparency. 

Response: 

- The assumptions and the model fitness t-test and plot provided as supplementary file 

Submit the STOBE checklist as a supplementary file. 

Response 

- STOBE checklist provided as supplementary file

---

## [Decision Letter · Decision Letter 4]

12 Nov 2023

PONE-D-22-15821R4Health providers readiness in managing intimate partner violence in public health institutions, Ethiopia.PLOS ONE

Dear Dr. Deribe,

Thank you for submitting your manuscript to PLOS ONE. After careful consideration, we feel that it has merit but does not fully meet PLOS ONE’s publication criteria as it currently stands. Therefore, we invite you to submit a revised version of the manuscript that addresses the points raised during the review process.

We look forward to receiving your revised manuscript.

Kind regards,

Humayun Kabir

Academic Editor

PLOS ONE

Journal Requirements:

Reviewers' comments:

Reviewer's Responses to Questions

**Comments to the Author**

1. If the authors have adequately addressed your comments raised in a previous round of review and you feel that this manuscript is now acceptable for publication, you may indicate that here to bypass the “Comments to the Author” section, enter your conflict of interest statement in the “Confidential to Editor” section, and submit your "Accept" recommendation.

Reviewer #1: All comments have been addressed

Reviewer #2: All comments have been addressed

2. Is the manuscript technically sound, and do the data support the conclusions?

Reviewer #1: Yes

Reviewer #2: Yes

3. Has the statistical analysis been performed appropriately and rigorously? 

Reviewer #1: Yes

Reviewer #2: Yes

4. Have the authors made all data underlying the findings in their manuscript fully available?

Reviewer #1: Yes

Reviewer #2: Yes

5. Is the manuscript presented in an intelligible fashion and written in standard English?

Reviewer #1: Yes

Reviewer #2: Yes

6. Review Comments to the Author

Reviewer #1: The manuscript is much stronger. One final suggestion is to add to the study objective the regression analysis - add that you are also examining factors associated with providers' perceived readiness to manage IPV

Reviewer #2: This interesting research examines health provider’s readiness to address IPV in Ethiopia. This topic is clearly of relevance to PLOS-One. The authors have responded conscientiously to a number of the reviewers’ comments and suggestions.

Formatting: OK.

Writing: Better

Title: Better.

Abstract: Better

Introduction: Better.

Research Methods: As before, the PREMIS tool is a strong choice. It is confusing to present the study variables separately and first if they are directly derived from the PREMIS (which seems the case). Present the PREMIS information first – then describe the subscales (which are the study variables) Consolidate this information.

Results: As before, this is a great sample size (N = 428).

Discussion: The comparison of the results to other PREMIS studies, especially other African studies, is interesting and appropriate.

Summary: In summary and as before, this Ethiopian study of IPV readiness with a large sample size using a well-respected tool, the PREMIS, has much to recommend it.

References: Correctly formatted.

7. PLOS authors have the option to publish the peer review history of their article (what does this mean?). If published, this will include your full peer review and any attached files.

Reviewer #1: No

Reviewer #2: No

---

## [Author Response · Author response to Decision Letter 4]

13 Nov 2023

We responded for all comments provided.

Response 1: objective related to regression analysis added in 

- abstract section introduction last sentence 

- page four line 70 and 72

Response 2: based on recommendation the study variables section is consolidated to the measurement section of methods.

---

## [Decision Letter · Decision Letter 5]

27 Nov 2023

Health providers readiness in managing intimate partner violence in public health institutions, Ethiopia.

PONE-D-22-15821R5

Dear Dr. Deribe,

We’re pleased to inform you that your manuscript has been judged scientifically suitable for publication and will be formally accepted for publication once it meets all outstanding technical requirements.

Kind regards,

Humayun Kabir

Academic Editor

PLOS ONE

Additional Editor Comments (optional):

Reviewers' comments:

Reviewer's Responses to Questions

**Comments to the Author**

1. If the authors have adequately addressed your comments raised in a previous round of review and you feel that this manuscript is now acceptable for publication, you may indicate that here to bypass the “Comments to the Author” section, enter your conflict of interest statement in the “Confidential to Editor” section, and submit your "Accept" recommendation.

Reviewer #1: All comments have been addressed

Reviewer #2: All comments have been addressed

2. Is the manuscript technically sound, and do the data support the conclusions?

Reviewer #1: Yes

Reviewer #2: Yes

3. Has the statistical analysis been performed appropriately and rigorously? 

Reviewer #1: Yes

Reviewer #2: Yes

4. Have the authors made all data underlying the findings in their manuscript fully available?

Reviewer #1: Yes

Reviewer #2: Yes

5. Is the manuscript presented in an intelligible fashion and written in standard English?

Reviewer #1: Yes

Reviewer #2: Yes

6. Review Comments to the Author

Reviewer #1: All feedback has been adequately addressed and I have no further feedback. This is a fifth revision and there are no further review comments.

Reviewer #2: As before, this interesting research examines health provider’s readiness to address IPV in Ethiopia. This topic is clearly of relevance to PLOS-One. The authors have responded conscientiously to a number of the reviewers’ comments and suggestions.

Formatting: OK.

Writing: Better

Title: Better.

Abstract: Better

Introduction: Better.

Research Methods: As before, the PREMIS tool is a strong choice. It is confusing to present the study variables separately and first if they are directly derived from the PREMIS (which seems the case). Present the PREMIS information first – then describe the subscales (which are the study variables) Consolidate this information.

Results: As before, this is a great sample size (N = 428).

Discussion: The comparison of the results to other PREMIS studies, especially other African studies, is interesting and appropriate.

Summary: In summary and as before, this Ethiopian study of IPV readiness with a large sample size using a well-respected tool, the PREMIS, has much to recommend it.

References: Correctly formatted.

7. PLOS authors have the option to publish the peer review history of their article (what does this mean?). If published, this will include your full peer review and any attached files.

Reviewer #1: No

Reviewer #2: No

---

## [Editor Report · Acceptance letter]

13 Dec 2023

PONE-D-22-15821R5 

PLOS ONE

Dear Dr. Deribe, 

I'm pleased to inform you that your manuscript has been deemed suitable for publication in PLOS ONE. Congratulations! Your manuscript is now being handed over to our production team.

Kind regards, 

on behalf of

Dr. Humayun Kabir 

Academic Editor

PLOS ONE